# 2D Nanomaterial, Ti_3_C_2_ MXene-Based Sensor to Guide Lung Cancer Therapy and Management [note 1]

**DOI:** 10.3390/bios11020040

**Published:** 2021-02-04

**Authors:** Mahek Sadiq, Lizhi Pang, Michael Johnson, Venkatachalem Sathish, Qifeng Zhang, Danling Wang

**Affiliations:** 1Biomedical Engineering Program, North Dakota State University, Fargo, ND 58108, USA; mahek.sadiq@ndsu.edu; 2Department of Pharmaceutical Science, North Dakota State University, Fargo, ND 58108, USA; lizhi.pang@ndsu.edu (L.P.); s.venkatachalem@ndsu.edu (V.S.); 3Materials and Nanotechnology Program, North Dakota State University, Fargo, ND 58108, USA; michael.johnson.1@ndsu.edu (M.J.); qifeng.zhang@ndsu.edu (Q.Z.); 4Department of Electrical and Computer Engineering, North Dakota State University, Fargo, ND 58102, USA

**Keywords:** 2D Ti_3_C_2_ MXene, PGE_2_, 8-HOA, lung cancer

## Abstract

Major advances in cancer control can be greatly aided by early diagnosis and effective treatment in its pre-invasive state. Lung cancer (small cell and non-small cell) is a leading cause of cancer-related deaths among both men and women around the world. A lot of research attention has been directed toward diagnosing and treating lung cancer. A common method of lung cancer treatment is based on COX-2 (cyclooxygenase-2) inhibitors. This is because COX-2 is commonly overexpressed in lung cancer and also the abundance of its enzymatic product prostaglandin E2 (PGE_2_). Instead of using traditional COX-2 inhibitors to treat lung cancer, here, we introduce a new anti-cancer strategy recently developed for lung cancer treatment. It adopts more abundant omega-6 (ω-6) fatty acids such as dihomo-γ-linolenic acid (DGLA) in the daily diet and the commonly high levels of COX-2 expressed in lung cancer to promote the formation of 8-hydroxyoctanoic acid (8-HOA) through a new delta-5-desaturase (D5Di) inhibitor. The D5Di does not only limit the metabolic product, PGE_2,_ but also promote the COX-2 catalyzed DGLA peroxidation to form 8-HOA, a novel anti-cancer free radical byproduct. Therefore, the measurement of the PGE_2_ and 8-HOA levels in cancer cells can be an effective method to treat lung cancer by providing in-time guidance. In this paper, we mainly report on a novel sensor, which is based on a newly developed functionalized nanomaterial, 2-dimensional nanosheets, or Ti_3_C_2_ MXene. The preliminary results have proven to sensitively, selectively, precisely, and effectively detect PGE_2_ and 8-HOA in A549 lung cancer cells. The capability of the sensor to detect trace level 8-HOA in A549 has been verified in comparison with the traditional gas chromatography–mass spectrometry (GC–MS) method. The sensing principle could be due to the unique structure and material property of Ti_3_C_2_ MXene: a multilayered structure and extremely large surface area, metallic conductivity, and ease and versatility in surface modification. All these make the Ti_3_C_2_ MXene-based sensor selectively adsorb 8-HOA molecules through effective charge transfer and lead to a measurable change in the conductivity of the material with a high signal-to-noise ratio and excellent sensitivity.

## 1. Introduction

The most common cancers occur in the lungs, breasts, pancreas, colon, skin, and stomach [1]. Lung cancer is the second most common cancer in men and women and the leading cause of cancer deaths in the United States. The two major types of lung cancer are small cell lung cancer (SCLC, ~15%) [2] and non-small cell lung cancer (NSCLC, ~85%) [3]. The survival rate of both types of lung cancer is very low [4]. According to the American Cancer Society, lung cancer and asbestos-related lung cancer [5] alone were responsible for 142,670 estimated deaths in 2019, making it the number one killer and three times deadlier than breast cancer [6]. This is because most patients (~75%) are diagnosed at a late stage of the disease (stage III or IV) [7]. To increase the survival rate, major advances in lung cancer control or prevention can be facilitated by early detection and effective anti-cancer therapy. In recent years, a variety of therapeutic and adjuvant methods and nutritional approaches have been developed for lung cancer treatment such as chemotherapy, targeted therapy [8,9], cyclooxygenase (COX)-2 inhibition [10], and omega-3 fatty acid dietary manipulation [11,12].

In addition to these methods, many physical “visualization/detection” methods [13] are available for tumor detection and cancer diagnosis [14]. Some of them are positron emission tomography (PET), magnetic resonance imaging (MRI), computerized tomography (CT), ultrasonography, endoscopy, and the gas chromatography method. However, these methods have some major issues for applications in cancer diagnosis. For example, MRI is very expensive and time-consuming. Sometimes it cannot even distinguish between malignant and benign cancer [15]. In the case of PET, radioactive material is used which is combined with glucose and injected into the patient. This process might cause a health concern for diabetic patients [16]. High-dose radiation involved in CT scanning can even increase the risk of cancer [17]. Ultrasound, however, cannot provide accurate diagnosis and frequently has difficulty determining whether a mass is malignant or not [18]. Endoscopy is relatively safer but still has complications such as perforation, infection, bleeding, and pancreatitis [19]. The fundamental limitation of gas chromatography is that the substance must be volatile. It means that a finite portion of the substance needs to be distributed into the gaseous state [20], which could make it problematic to use gas chromatography–mass spectroscopy (GC–MS) in cancer detection because its sampling procedure is very complicated and the results are difficult to interpret. In addition, the GC–MS technique is very expensive and must be operated by very skilled personnel [21]. Therefore, an effective and accurate technique to diagnose cancer and assist in effective treatment is urgently needed.

In particular, studies have confirmed that cyclooxygenase (COX), typically the inducible form of COX-2, is commonly overexpressed in lung cancer and the abundance of its enzymatic product prostaglandin E_2_ (PGE_2_) plays an important role in influencing cancer development. Since PGE_2_ is a deleterious metabolite formed from COX-2-catalyzed peroxidation of an upstream omega-6 (ω-6) fatty acid called arachidonic acid (AA), PGE_2_ promotes tumor growth and metastasis [22]. Subsequently, it can be treated as an indicator of local COX activity to regulate or control lung cancer. Many efforts in lung cancer therapy have been focused on the development of COX-2 inhibitors because they can be used to suppress prostaglandinE_2_ (PGE_2_) formation from COX-2-catalyzed ω-6 arachidonic acid peroxidation [23]. However, most COX-2 inhibitors can severely injure the gastrointestinal tract, increase the risk of cardiovascular disease, and provide limited clinical responses [22,23]. To seek a safer and more efficient method to treat cancers, a new anti-cancer strategy [24], as shown in Figure 1, has been recently developed. This is a very different approach than the classic COX-2 inhibitors [24,25,26]. In detail, this is a strategy that adopts more abundant ω-6s such as dihomo-γ-linolenic acid (DGLA) in the daily diet and the commonly high level of COX expressed in most cancers to promote the formation of 8-hydroxyoctanoic acid (8-HOA) using a newly developed inhibitor, delta-5-desaturase inhibitor (D5Di). This is because the D5D is an enzyme that converts an upstream DGLA in a diet to AA. The high expression of COX-2 will promote the conversion of AA to PGE_2_, while the D5Di will (1) knock down the conversion of DGLA to AA and limit the generation of metabolic product, PGE_2_; and (2) promote the COX-2-catalyzed DGLA peroxidation to form 8-HOA, a novel anti-cancer free radical by-product. This strategy has proven to produce more effective and safer therapeutic outcomes in cancer treatment and has been validated in colon and pancreatic cancers [27]. Therefore, detection of the PGE_2_ and 8-HOA in lung cancer should be an effective method to evaluate the efficiency of the cancer treatment. Furthermore, the relative ratio of PGE_2_ and 8-HOA concentrations can become a useful adjuvant method to help diagnose cancers at an early stage. Therefore, it is very critical to develop a sensing technique or device that can track/monitor the PGE_2_ and 8-HOA concentrations in cancer and provide in-time guidance and feedback for cancer treatment and prevention. However, due to the extremely low concentrations of PGE_2_ and 8-HOA in cancer cells ~ ng/mL or µM, the detection of these components is quite challenging. Traditional methods of measuring low concentrations of compounds, such as PGE_2_ and 8-HOA, are using gas chromatography–mass spectrometry (GC–MS) or liquid chromatography–mass spectrometry (LC–MS). These techniques, as described above, are accurate and sensitive but heavy (not portable), expensive (needing skilled personnel to operate), and time-consuming (complicated sample processing) and cannot provide in-time feedback.

Recent advances in nanotechnology have made it possible to synthesize functionalized nanomaterials for applications in electronics, sensing, biomedicines for disease diagnosis and control, drug delivery, and the food industry [28,29,30,31]. Due to the increased surface areas and the feasibility of controllable size and surface properties, nanomaterials such as nanofibers, nanowires, and nanoparticles provide great opportunities for the development of advanced sensing systems and portable device/instrumentation with improved sensitivity and selectivity [32,33,34,35,36,37,38]. In particular, the use of structure-directing synthetic approaches in nanomaterials allows the tailoring of the nanomaterial crystalline phase, surface states, morphology, and facets for specific sensing application. Recently, with the development of two-dimensional (2D) nanomaterials such as graphene, these types of materials have gained tremendous attention because of their astonishing electrical and optical properties featured with an “all-surface” nature [39,40,41,42,43]. This all-surface nature can offer great opportunities to tune material properties through surface treatment for targetable detection. 

In 2011 [44], the discovery of MXenes introduced a new family into the two-dimensional (2D) materials and further proves to be promising in the flexible and broad application due to its controllable preparation methods and fascinating properties. In essence, MXenes consist of transition metals (including Ti, V, Nb, Mo, etc.) and carbon or nitrogen, sharing a general formula of *M_n_*_+1_*X_n_* (*n* = 1–3). As a new star of 2D materials, MXenes have the metallic conductivity and hydrophilic nature due to their uncommon surface terminations. Moreover, the unique accordion-like morphology (Figure 2), excellent conductivity, and rich but tailorable surface functional groups endow MXenes with attractive electronic, mechanical, physical, and chemical properties [45] for applications in energy storage [46], environmental science [47], and sensors [48]. The numerous applications of MXene as sensors in various fields are summarized in the paper [49]. In our conference paper [50], we reported that the as-synthesized Ti_3_C_2_ MXene-based nanosensor has effectively detected 8-HOA in cancer cells without and with using a D5D inhibitor. In that paper, we described more details about how the new sensor based on the two-dimensional nanomaterial Ti_3_C_2_ MXene [51] can facilitate the lung diagnosis and treatment efficiently by using the new D5D inhibitor and 8-HOA therapy on lung cancer. The preliminary data indicate that this new sensor device can sensitively detect PGE_2_ and 8-HOA levels in healthy and cancerous lung cells (BEAS2B and A549 respectively) with similar accuracy to GC–MS but much faster and in an in-time manner to guide the cancer treatment.

## 2. Materials and Methods

### 2.1. Sensing Material Synthesis, Sensing Tests, and Cell Lines Preparation

#### 2.1.1. Ti_3_C_2_ Nanomaterial-Based Sensor Preparation

The sensor that we used is based on a new 2D nanomaterial, Ti_3_C_2_ MXene. This nanomaterial was prepared using a method developed in our group and named the “hot etching method” [52]. In detail, the synthesis of Ti_3_C_2_ MXene followed the steps: (1) Preparing the Ti_3_AlC_2_ MAX phase. It was obtained through ball milling TiC, Ti, and Al powders in the molar ratio 2:1:1.2 respectively, for 5 h. Under argon flow, the resulting powder was then pressed into a pellet and sintered at 1350 °C for 4 h. The collected pellet after being milled back into powder was sieved through a 160-mesh sieve; (2) etching Al from the MAX phase to form the MXene phase. The as-prepared MAX powder was collected at an elevated temperature for etching through the “hot-etch method”. Hydrofluoric (HF) acid in 25 mL Teflon line autoclave at a temperature of 150 °C was used in a Thermolyne furnace for 5 h to etch 0.5 g of the MAX phase. To remove Al from the MAX phase, 5%wt of HF was used. Materials after being sonicated for one hour using a sonicating bath were collected through centrifuge; all the materials were dried overnight in a drying oven at 65 °C; (3) synthesizing MXene powders for the sensing film. Finally, the synthesized nanomaterial was drop-casted on the gold electrode-patterned glass substrate to form a thin film. The thin film is made by first making a paste for the application of the MXene powder to the substrate. This paste is made using 0.1 g of MXene material in 0.3 mL of ethanol which is then dispersed via mixing on a stir-plate. This paste is then blade-coated onto the sensor substrate at a thickness of 0.05 mm. The morphology of the synthesized 2D multilayered nanomaterial is shown in Figure 2a as the scanning electron microscope (SEM) image, which clearly exhibits multilayered nanosheets and accordion-like morphology. Figure 2b reveals the as-synthesized Ti_3_C_2_ material’s special surface terminations, which can lead to the unique surface and material properties of Ti_3_C_2_.

#### 2.1.2. Ti_3_C_2_ MXene Based Sensor Device Fabrication

In order to monitor the promoting 8-HOA formation and the variation of PGE_2_ in cancer cells during the new anti-cancer treatment, the sensor device, as shown in Figure 3, is fabricated followed by the steps: (1) Ti_3_C_2_ MXene sensing film fabrication: the Au electron contact patterned using photolithography and deposited on the glass wafer as the substrate, and then direct drop-casting the as-synthesized Ti_3_C_2_ suspension solution onto the patterned substrate to form the sensor slide; (2) the resistance change caused by the exposure of 8-HOA/PGE_2_ on the sensing film is measured through a Keithley resistance meter and data collected via computer.

#### 2.1.3. Cancer Cell Lines and Materials

A549 (ATCC^®^CCL-185™), NCI-H1299 (ATCC^®^ CRL-5803), and BEAS-2B (ATCC^®^CRL-9609™) were purchased from American Type Culture Collection (ATCC, VA, USA). Iminodibenzyl (CAS Number: 494-19-9) and 8-hydroxyoctanoic acid (8-HOA) were obtained from Sigma-Aldrich (St. Louis, MO, USA). PGE_2_ and DGLA (for in vitro study) and DGLA ethyl ester (for in vivo study) were acquired from Cayman Chemical (Ann Arbor, MI, USA).

#### 2.1.4. Preparation of Cell Samples

About 3 × 10^5^ A549 or BEAS-2B cells were trypsinized and seeded into each well of the 6-well plates. Then, the cells were randomly assigned into different groups for the administration of DGLA (100 μM), iminodibenzyl (10 μM), or their combination accordingly. After 48 h, the cell culture medium was collected. Cells were washed with phosphate buffer solution (PBS) and collected by centrifugation after trypsinization. A 1 mL cell culture medium with collected cells was homogenized and ready for testing. Three different groups of control samples were prepared using the same preparation procedures, including (a) blank group in 1 mL cell homogenate without any treatment; (b) 8-HOA group in 1 mL cell homogenate containing 0.6 ug/mL exogenous 8-HOA; (c) PGE_2_ group in 1 mL cell homogenate containing 6 ug/mL exogenous PGE_2_.

#### 2.1.5. Xenografted Lung Tumor Model on Nude Mice

Six-week-old nude mice were purchased from The Jackson Laboratory. The mice were housed in a pathogen-free IVC System with water and food ad libitum. All the animal experiments in this study were approved by the Institutional Animal Care and Use Committees at North Dakota State University. About 2 × 10^6^ A549 or H1299 cells were injected into the hind flank of the nude mouse to induce tumors as we previously described [26]. The mice were randomly assigned to the following treatments: Control (treated with the same volume of the vehicle), DGLA (5 mg/mouse, oral gavage, every day), iminodibenzyl (15 mg/kg, intraperitoneal injection, daily), and DGLA+ iminodibenzyl. The treatment started at two weeks of injection of A549 cells in nude mice. All the administrations lasted for four weeks. At the end of the treatment, mice were sacrificed, and tumors were isolated. Tumor tissues were crushed and homogenized by using a mortar in liquid nitrogen. The blood was centrifuged for 10 min at 2000 rpm for separating serum. The supernatant of tumor tissues and serum was collected for analysis.

### 2.2. Methodology

To verify the roles of 8-HOA and PGE_2_ in cancer development and treatment, the experiments have been designed to do testing in healthy lung cells and A549 lung cancer cells. The 8-HOA and PGE_2_ concentrations were carefully calculated before applying them onto the sensing films with certain accuracy and consistency. The sensor performance was further verified in comparison with the measurement using traditional GC–MS.

#### 2.2.1. Normal Cells

For application to test the effect of 8-HOA and PGE_2_ in normal lung cells, 10^6^ BEAS2B non-tumorigenic epithelial cell lines were collected. 8-HOA, PGE_2_, and BSA (Bovine Serum Albumin) were applied to the samples right before measuring the resistance change. Once the samples were applied onto the Ti_3_C_2_ MXene-based sensors, resistances were measured immediately and repeated at regular time intervals. The experiment is listed in Table 1 and the resistance change of the MXene slides for each of the samples is measured and shown in Figure 4. The resistance increases dramatically when BEAS2B is added with PGE_2_ but BEAS2B alone and BEAS2B with 8-HOA do not show obvious change of resistance.

#### 2.2.2. A549/H1299 Lung Cancer Cells

A549 and H1299 both are lung cancer epithelial cell lines. To study the effect of 8-HOA, both H1299 and A549 cell lines have been studied on lung cancer cell apoptosis, proliferation, and survival as shown in Figure 5. H1299 cells were treated with 1 μM 8-HOA for 48 h and afterward were subjected to flow cytometry to observe the apoptosis in the staining of Annexin V-FITC/PI. Wound-healing assay of H1299 lung cancer (non-small cell lung carcinoma cell line) treated with 8-HOA was observed and recorded having the relative area of the wound to 0 **h** respectively normalized to 1. Finally, the Colony formation assay of H1299 lung cancer cells treated with 8-HOA was observed. Survival fraction of different treatment groups to control were respectively normalized to 1.

Figure 6 shows the effect of D5D inhibitor on lung cancer apoptosis. Cell apoptosis was determined by flow cytometry on H1299 lung cancer cells in staining of Annexin V-FITC/PI. H1299 cells were treated with DGLA (100 μM) and D5D inhibitor (10 μM) for 48 h before flow cytometry analysis. Immunofluorescence images of cleaved poly (ADP-ribose) polymerase (PARP) in lung tumor tissues after 4 weeks of treatment with DGLA (5 mg/mouse) and D5D inhibitor (15 mg/kg) were collected. The expression of cleaved PARP was stained in violet, and cell nuclei were counter-stained with 4′,6-diamidino-2-phenylindole (DAPI).

The experimental results showed a similar response to 8-HOA in A549 and H1299 cell lines. In sensing tests, A549 cells were collected after being cultured. The complete design of the experiments to verify the relative concentration of generated 8-HOA and PGE_2_ with and without using the new cancer treatment via the detection of Ti_3_C_2_ Mxene-based sensor are listed in Table 2. Similar to the BEAS-2B cell lines, 8-HOA and PGE_2_ samples were applied to the A549 cell lines just before conducting the experiment. The sensing tests of these samples are shown in Figure 7. The resistances of A549 cancer cells, A549 cells treated by DGLA, and PGE_2_ are much higher than the cancer cells treated by adding 8-HOA, applying D5Di, or using the new anti-cancer treatment DGLA + D5Di.

## 3. Results and Discussion

### 3.1. Observation from the Non-Tumorigenic Sample Graph

In a healthy subject, both the concentration of PGE_2_ and 8-HOA should be low. The sensing test is conducted on the normal lung cell, BEAS2B, without extra treatment and BEAS2B by treating with extra PGE_2_ or 8-HOA. A significant resistance increase is observed in BEAS2B by adding 10 µM PGE_2,_ while the untreated normal cells and cells treated by 8-HOA do not show obvious resistance change. This result indicates a unique role of PGE_2_ in healthy cells through the change of the electrical property of sensing material. Considering the elevated concentration of PGE_2_ can indicate a cancer development, such a sensitive response to PGE_2_ using Ti_3_C_2_ MXene-based sensor can be potentially used to diagnose cancer even at a very early stage.

### 3.2. Observation from the CARCINOGENIC Samples

As we have discussed in this paper previously, D5D inhibitor (D5Di) is used for preventing the conversion of DGLA to AA and ultimately limiting the formation of PGE_2._ According to the main mechanism of the new anti-cancer strategy, D5Di along with DGLA can effectively limit the formation of PGE_2_ but promote the formation of 8-HOA. The sensing test using the newly developed Ti_3_C_2_ MXene-based sensor, as shown in Figure 6, exhibits an interesting trend of resistance change. Similar to showing high resistance for A549, A549 with adding 10 uM PGE_2_, and A549 treated by DGLA both show high resistance in the sensing test. The results indicate a higher concentration of PGE_2_ generated in A549 cells while the high resistance in the sample only treated by DGLA confirms that omega-6 (DGLA) are pro-inflammatory and promote the formation of PGE_2_. However, the new anti-cancer treatment using DGLA and D5Di to treat A549 cells shows a similar low resistance level to that of A549 cells with 8-HOA. This result indicates promising information: the Ti_3_C_2_ MXene-based sensor can be used to monitor or validate the anti-cancer effect of the new strategy: DGLA + D5Di, which should be an effective anti-cancer effect because of the generation of 8-HOA.

### 3.3. Correlation between the Sensing Test Results and GC–MS Results

To verify the Ti_3_C_2_-based sensor for PGE_2_ and 8-HOA detection, both sensor and GC–MS have been used to detect very low concentrations of 8-HOA and PGE_2_ in A495 lung cancer cells. The Ti_3_C_2_ Mxene sensors can provide the information of concentration of 8-HOA via the value of resistance while GC–MS can quantitatively provide the exact concentration of 8-HOA. As shown in Figure 8, an obvious correlation is obtained between the GC–MS measurement and resistances that the Ti_3_C_2_ MXene-based sensor measured. This correlation further confirmed the capability of the Ti_3_C_2_ MXene-based sensor to detect trace concentrations of 8-HOA. It can be a convenient, fast, and low-cost tool to help the anti-cancer strategy in lung cancer treatment.

## 4. Conclusions and Discussion

A new sensor based on 2D nanosheets, Ti_3_C_2_ MXene, has been designed and used for the sensing response to 8-HOA and PGE_2_ in lung cancer cells. The preliminary results indicate an important conclusion: this new Ti_3_C_2_-based sensor can provide a convenient and simple method for anti-cancer treatment guidance. In addition, the high sensitivity of this new sensor opens a potential application for early-stage cancer detection via monitoring variation of PGE_2_ and 8-HOA in cells. Instead of using heavy, expensive, and time-consuming GC–MS to assist the anti-cancer treatment, the Ti_3_C_2_ MXene-based sensor can provide a fast, simple, low-cost, highly efficient, and much less invasive assistant tool to detect and cure cancer.

## Figures and Tables

**Figure 1 biosensors-11-00040-f001:**
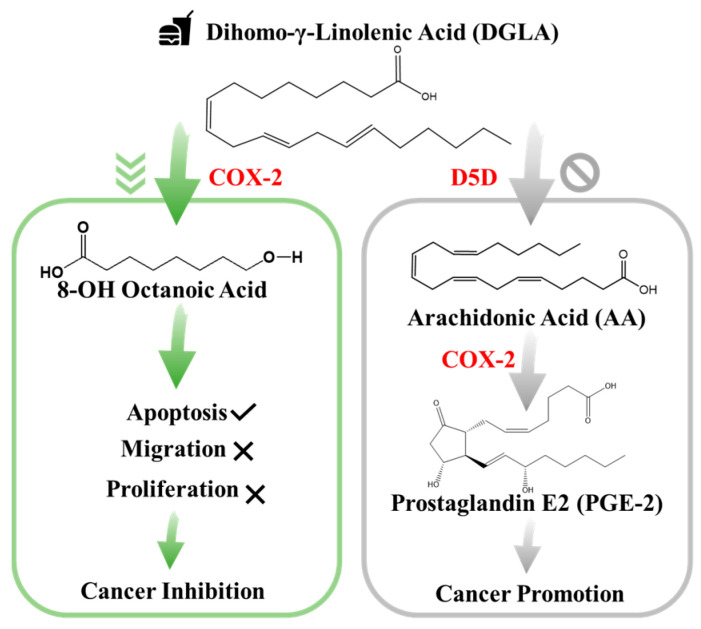
New anti-cancer strategy: target but do not inhibit cyclooxygenase-2 (COX-2) in cancer.

**Figure 2 biosensors-11-00040-f002:**
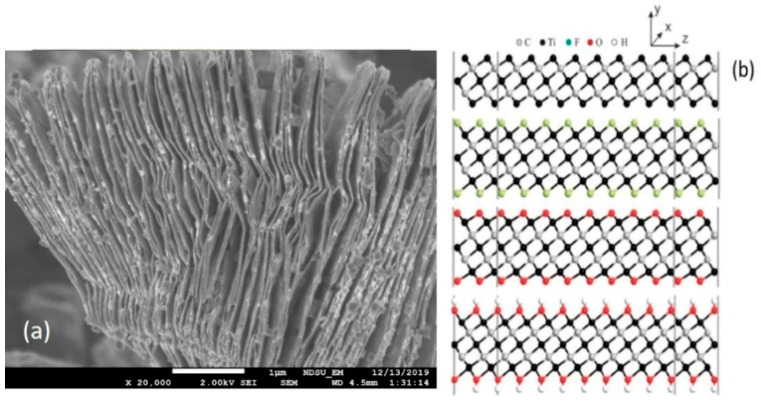
Newly synthesized 2D multilayered Ti_3_C_2_ MXene nanosheets (**a**) Scanning electron microscope (SEM) image; (**b**) pristine and surface-terminated Ti_3_C_2_ MXene with different functional groups.

**Figure 3 biosensors-11-00040-f003:**
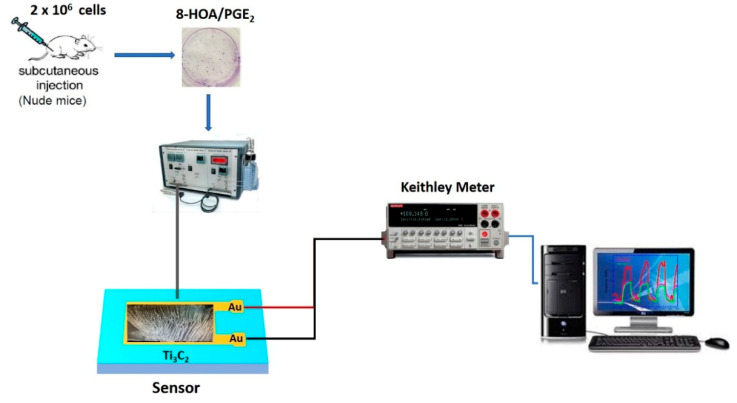
Sketch of Ti_3_C_2_ MXene-based sensing system.

**Figure 4 biosensors-11-00040-f004:**
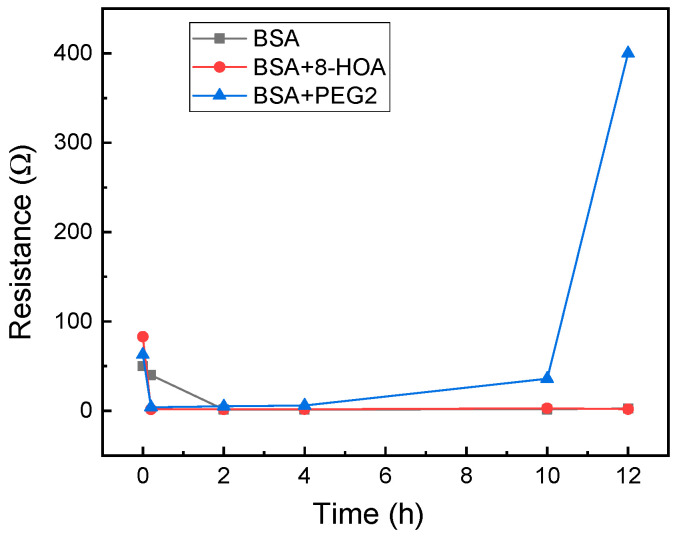
Resistance change measured using Ti_3_C_2_ MXene-based sensors for BEAS2B cells.

**Figure 5 biosensors-11-00040-f005:**
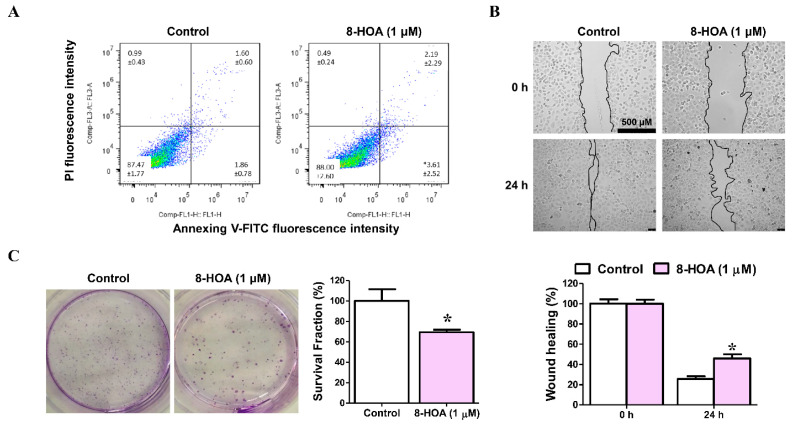
Effect of 8-hydroxyoctanoic acid (8-HOA) on lung cancer cell apoptosis, proliferation, and survival. (**A**) Cell apoptosis was determined by flow cytometry. (**B**) Wound-healing assay of H1299 lung cancer cells. (**C**) Colony formation assay of H1299 lung cancer. * *p* < 0.05 vs. Control group. Data represent mean ± SEM, unpaired t-test.

**Figure 6 biosensors-11-00040-f006:**
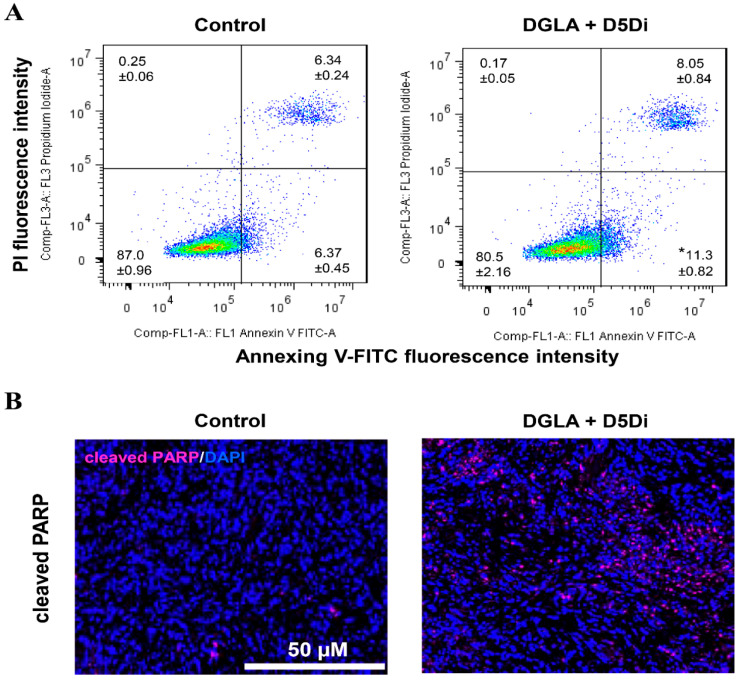
Effect of delta-5-desaturase (D5D) inhibitor on lung cancer apoptosis. (**A**) Cell apoptosis determined by flow cytometry. (**B**) Immunofluorescence images of cleaved poly (ADP-ribose) polymerase (PARP) in lung tumor tissues. * *p* < 0.05 vs. Control group. Data represent mean ± SEM, unpaired t-test.

**Figure 7 biosensors-11-00040-f007:**
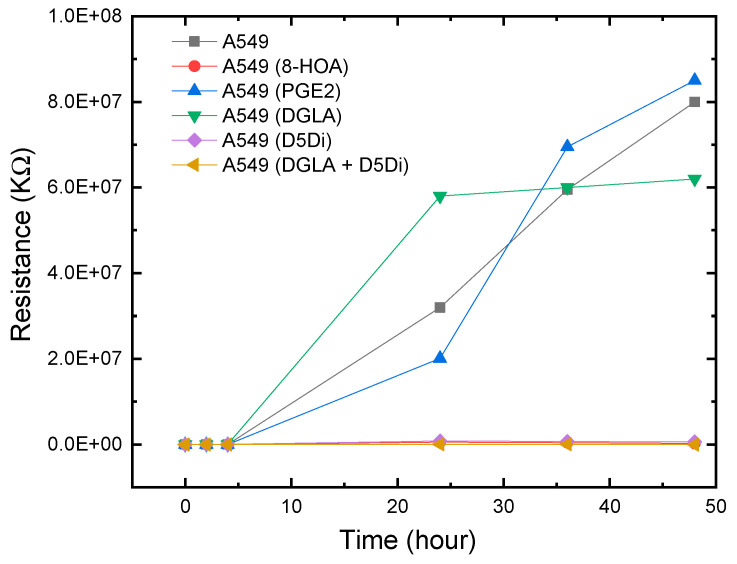
Resistance change measured using Ti_3_C_2_ MXene-based sensors for A549 cancer cells with and without using the new anti-cancer treatment.

**Figure 8 biosensors-11-00040-f008:**
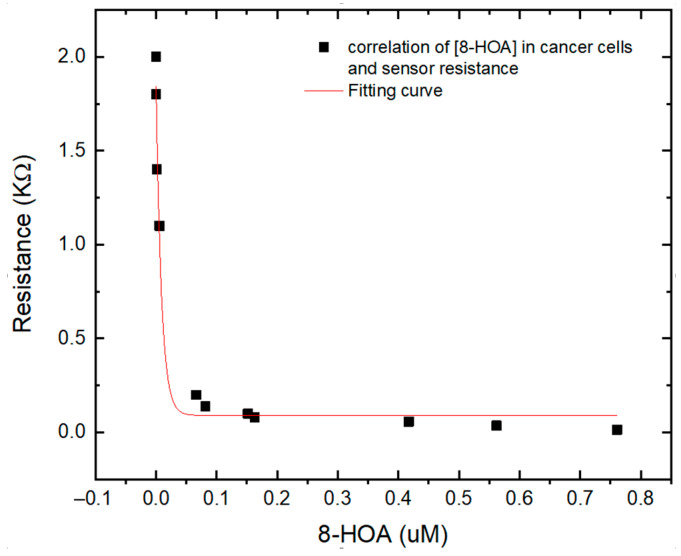
Correlation between different concentration of 8-HOA detected by gas chromatography–mass spectroscopy (GC–MS) and resistance measured by Ti_3_C_2_ MXene sensor using the same sampling conditions.

**Table 1 biosensors-11-00040-t001:** Table showing the composition of each sample for BEAS2B.

Sample	Cell	8-HOA	PGE_2_	BSA
1	10^6^ BEAS2B	none	none	None
2	10^6^ BEAS2B	0.6 ug/mL	none	None
3	10^6^ BEAS2B	none	6 ug/mL	None
4	None	none	none	1 mg/mL

**Table 2 biosensors-11-00040-t002:** The composition of each sample for A549 cells treated by 8-hydroxyoctanoic acid (8-HOA), Prostaglandin E2 (PGE2), dihomo-γ-linolenic acid (DGLA), delta-5-desaturase inhibitor (D5Di), and DGLA + D5Di.

Sample	Cell	DGLA	D5Di	8-HOA	PGE_2_	Estimated 8-HOA/PGE_2_ Level
1	10^6^ A549	none	none	none	none	Low 8-HOA; low PGE2
2	10^6^ A549	none	none	0.6 μg/mL	none	High 8-HOA; low PGE2
3	10^6^ A549	none	none	none	6 μg/mL	Low 8-HOA; high PGE2
4	10^6^ A549	100 μM	none	none	none	Low 8-HOA; high PGE2
5	10^6^ A549	none	10 μM	none	none	Low 8-HOA; low PGE2
6	10^6^ A549	100 μM	10 μM	none	none	High 8-HOA; low PGE2

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
