# Peer review of "2D Nanomaterial, Ti3C2 MXene-Based Sensor to Guide Lung Cancer Therapy and Managementâ€"

_biosensors, 2021, doi:10.3390/bios11020040_

Round 1
Reviewer 1 Report
This work has potential, however, need improvement based on the following comments.
- Line 137-148; Not much details such as amount (MXene aqueous dispersion) and layer thickness of fabricated sensor were mentioned. Authors should provide complete details.
- In reference to sensor, stability test is critical. Is there any leaching of MXene from sensor surface during experiments? Did author test it?
- Please mention Limit of detection (LOD) in the text.
- What about the selectivity of the sensor? Authors should mention and work in this area.
- Some recent MXene articles should be cited. i.e.
-2D Transition Metal Carbides (MXene) for Electrochemical Sensing: A Review,
- MXene: An emerging material for sensing and biosensing
Author Response
We would like thank you all for the valuable comments provided by the reviewer. We have revised the manuscript utilizing their comments, and rewritten paper to address the concerns and comments given by the reviewer. Below are our responses to the comments of the reviewer.
Reviewer 1
This work has potential, however, need improvement based on the following comments.
(1) Line 137-148; Not much details such as amount (MXene aqueous dispersion) and layer thickness of fabricated sensor were mentioned. Authors should provide complete details.
Response: Thank you for your concerns, this section has been further expanded upon to better explain the fabrication and dimensions of the finished sensor film. See lines 142-145.
(2) In reference to sensor, stability test is critical. Is there any leaching of MXene from sensor surface during experiments? Did author test it?
Response: Thank you for your helpful insight. Regarding the leaching issue, we did notice it and tried to avoid it via carefully adjusting the amount of samples on the MXene sensor surface (lines 185-187 in the revised paper). Each experiment, our collaborator (Lizhi Pang) has quantified the solution and no obvious leaching observed. The treated sensor slide should be consistent either from sensing material and film fabrication and concentration of sensing analyte.
(3) Please mention Limit of detection (LOD) in the text.
Response: Thank you for your feedback. Since the main purpose of the experiment does not care about how low the concentration of 8-HOA could be, it is enough to know that the sensor can sensitively and selective detect the variation of [8-HOA] and [PGE2] in normal cells and cancer cells. Therefore, the threshold concentration of 8-HOA and PGE2 is more important than the detection limit. However, we did perform the detection limit of sensor to 8-HOA, as shown in Figure 8, which is down to 0.0014 uM.
(4) What about the selectivity of the sensor? Authors should mention and work in this area.
Response: Thank you for your suggestion. First of all, it will be helpful to address your concern if you can provide more information about selectivity referring to. In our paper, Figure 7 has been providing the information of selectivity to detect 8-HOA and PGE2 at different conditions of cancer cells. This should indicate the specificity of the sensor.
(5) Some recent MXene articles should be cited. i.e.
-2D Transition Metal Carbides (MXene) for Electrochemical Sensing: A Review,
- MXene: An emerging material for sensing and biosensing
Response: Please refer the response in line 115 and lines 423-434 in our revised paper.
Reviewer 2 Report
This paper presents 2D nanomaterial, Ti3C2MXene based sensor to guide lung cancer therapy and management. Even though the article is interesting, I have the following concerns:
Introduction part should be supplemented about information connected with recent works on Ti3C2MXene, especially that presented in this paper sensor is based on it.
Please increase the quality and dimensions of Figures. Figures 2,5, 6 are to small, not clear and it is very difficult to recognize scale, numbers, etc. If these results are to confirm that the Ti3C2 MXene based sensor can be used to monitor or validate the anti-cancer effect, they should be clear.
Since the results presented in Tables 1 and 2 and Figures 4,7 have been published in Proceedings, 60(1), 29. doi:10.3390/iecb2020-07055, authors should show what new content or results were added to this paper.
Author Response
We would like thank you all for the valuable comments provided by the reviewer. We have revised the manuscript utilizing their comments, and rewritten paper to address the concerns and comments given by the reviewer. Below are our responses to the comments of the reviewer.
This paper presents 2D nanomaterial, Ti3C2MXene based sensor to guide lung cancer therapy and management. Even though the article is interesting, I have the following concerns:
- Introduction part should be supplemented about information connected with recent works on Ti3C2MXene, especially that presented in this paper sensor is based on it.
Response: Thanks a lot for your suggestions. Please refer to the lines 116-123 in the revised paper which focus on the connection and motivation why we choose Ti3C2 MXene as the sensing material to detect trace levels of 8-HOA and PGE2 in lung cancer treatment.
- Please increase the quality and dimensions of Figures. Figures 2, 5, 6 are to small, not clear and it is very difficult to recognize scale, numbers, etc. If these results are to confirm that the Ti3C2 MXene based sensor can be used to monitor or validate the anti-cancer effect, they should be clear.
Response: Thanks for suggestions. We replaced the SEM image with clear scale in the revised paper, page 3, line 126. The Figure 5 and Figure 6 were attached in the revised paper, pages 4 and 5.
- Since the results presented in Tables 1 and 2 and Figures 4,7 have been published in Proceedings, 60(1), 29. doi:10.3390/iecb2020-07055, authors should show what new content or results were added to this paper.
Response: Thanks a lot for your suggestions. This is a selected paper from the proceedings. However, we almost rewrote the whole section of “results and discussion” when we submitted the conference paper to biosensor journal. Specifically, we include new data to verify the sensor performance by comparing to the testing 8-HOA concentration using GC-MS (lines 186-187 on page 5). Also, we added Figures 5, 6 to show the effect of D5D inhibitor and its byproduct 8-HOA on lung cancer. We observed that both D5D inhibitor and 8-HOA significantly inhibited lung cancer growth in vitro (Figure 5 and 6A) and in vivo (Figure 6B) by regulating cancer cell apoptosis, migration, and proliferation. These results implicated the potential of D5D inhibitor and 8-HOA as a novel therapy on lung cancer, therefore, the Ti3C2 MXene based sensor we reported in this study could facilitate the diagnosis and expand the benefit of this treatment to lung cancer patients. Figure 8 has verified the effectivity of new sensor comparing to traditional method GC-MS.
Round 2
Reviewer 1 Report
No
Reviewer 2 Report
Authors have well addressed the reviewers' comments, the revised manuscript is thus acceptable for publication in Biosensors